# Small Muddy Paleochannels and Implications for Submarine Groundwater Discharge near Charleston, South Carolina, USA

Scott M. White * , Erin Smoak, Andrew L. Leier and Alicia M. Wilson

School of the Earth, Ocean, and Environment, University of South Carolina, Columbia, SC 29208, USA; esmoak@geol.sc.edu (E.S.); aleier@geol.sc.edu (A.L.L.); awilson@seoe.sc.edu (A.M.W.)
* Correspondence: swhite@geol.sc.edu; Tel.: +1-803-777-6304

**Abstract:** The spatial variations in Quaternary sediments on the inner continental shelf are produced by the progression of depositional environments during the latest sea-level rise, and this sedimentary architecture plays a fundamental role in controlling groundwater discharge. However, coincident seismic mapping, sediment cores, and hydrological studies are rare. Here, we combine high-resolution, 0.5–10 kHz, high-frequency seismic profiles with sediment cores to examine the nature of the sediment deposits, including paleochannels, where submarine groundwater discharge has also been studied in a 150 km$^2$ area of the inner shelf north of Charleston, South Carolina. We used high-frequency seismic reflection to interpret seismic facies boundaries, including 16 paleochannel crossings, to 20 km offshore. From 13 vibracores taken at the intersections of the seismic lines, we defined seven lithofacies representative of specific depositional environments. The paleochannels that we cored contain thick layers of structureless mud sometimes interbedded with silt, and mud is common in several of the nearshore cores. Our results indicate that paleochannels are often mud-lined or filled in this area and were most likely former estuarine channels. Neither the paleochannels nor a mud layer were found farther than 11 km off the present shoreline. This offshore distance coincides with the strongest pulses of groundwater discharge, emerging just beyond the paleochannels. This suggests that the muddy paleochannel system acts as a confining layer for submarine groundwater flow.

**Keywords:** continental shelf; CHIRP; depositional environments; coastal aquifer; Quaternary

## 1. Introduction

For decades, Quaternary sediments on continental shelves in passive margin settings have commonly been approached as archives of sedimentary environments over the last several glacial cycles, which create a complex and locally variable "mosaic" of sedimentary units seen on modern coastlines [1]. This mosaic of offshore sedimentary units contains records of past environments and affects the present availability of resources [2], including water resources [3]. In recent years, it has also become apparent that these sediments mitigate a significant exchange of seawater and groundwater. The discharge of fresh groundwater to the ocean can transmit contaminants and nutrients [4], influencing nearshore marine ecosystems and contributing to the eutrophication of estuaries and enclosed bays [5]. On a much larger scale, chemical tracer studies suggest that saline groundwater discharge may contribute as much water to the coastal ocean as river discharge, with important implications for the nutrient budgets in coastal seawater [6]. It is now apparent that the majority of this saline discharge occurs in pulses [7,8], and new observations show that this discharge can occur kilometers offshore, likely controlled by the sub-seafloor stratigraphic architecture [9]. Overall, several scales of groundwater flow and submarine groundwater discharge (SGD) have been recognized [10]. Very few field studies have addressed flows at the km scale, and even fewer have been able to link the underlying geology with the groundwater flow at the seafloor.

The primary way that continental shelf sedimentary geology has been connected with SGD is through paleochannels [11–13]. Paleochannels, which are common features

in continental shelves, are relict fluvial channels from sea-level lowstands [14], and they have also been suggested as reservoirs or conduits for sub-seafloor flow of freshened water [12,13,15–18]. Thus, paleochannels may put limits on the location of the freshwater–saltwater interface offshore in coastal aquifers, and hence the extent of fresh groundwater resources. Also, paleochannels may act as pathways for saltwater intrusion into these aquifers. The geochemical reactions and salinization of aquifers in general are key to their availability as a resource [19–29]. Offshore groundwater may become important as coastal development continues and sea levels rise [30,31]. Despite the potential importance of paleochannels in this context, few studies have sampled paleochannel and interfluvial sediments, and it is often assumed that the paleochannel fill is composed of sand and similar relatively coarse-grained sediment with high-porosity and permeability [11–13,15,17], which may not be valid for smaller paleochannel systems [32].

This study examines sediment deposits 2–20 km off the Isle of Palms near Charleston, South Carolina, using an integrated geophysical and sedimentological approach that included 13 sediment vibracores collected at intersections of high-resolution seismic profiles (Figure 1). To investigate depositional environments of the coast and the nature of paleochannels as possible pathways for SGD, we define the local near-surface stratigraphy and then compare the lithofacies, porosity, and permeability of paleochannel fill deposits to interfluvial sediments. The main goal of this study is to understand the role of the sedimentary architecture in SGD. Our results and the supporting SGD models [9] illustrate the importance of the geometry of shallow confined aquifers as a control on the spatial pattern of SGD and guide future possible sampling and monitoring efforts.

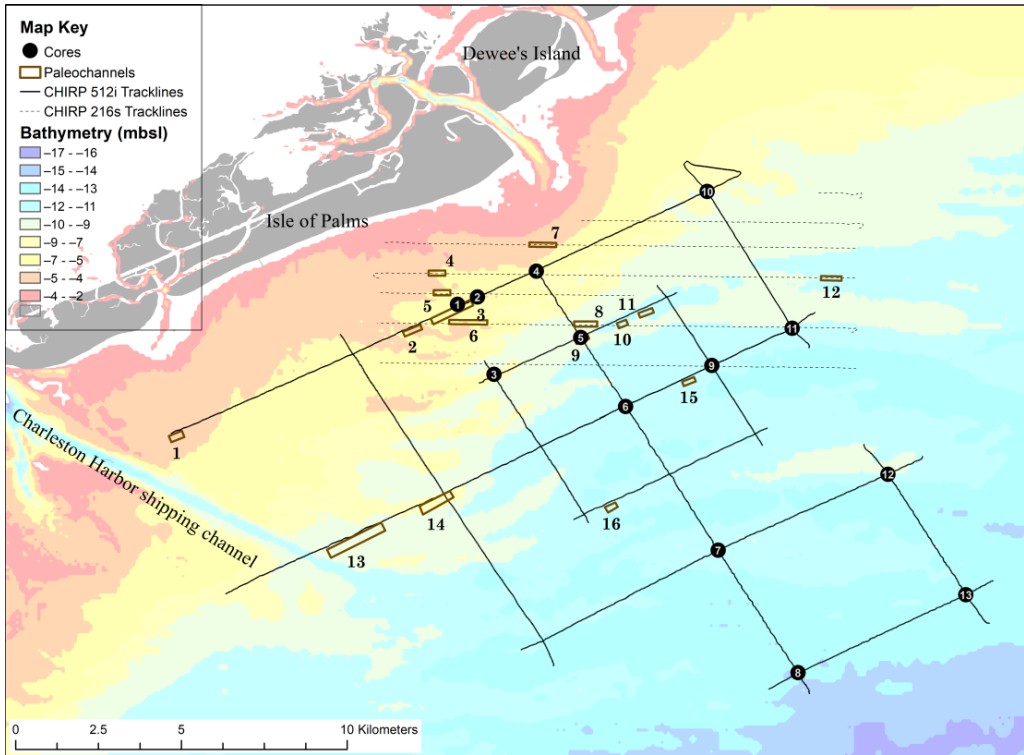

**Figure 1.** Bathymetry of the study site off Isle of Palms, South Carolina, with Dewee's Island and the entrance to Charleston Harbor and shipping channel shown for reference. Seismic (CHIRP) survey lines collected in this study (black lines) and from [33] (dashed). Sediment vibracores (black dots) are labeled with core number. Paleochannel crossings identified from the sub-bottom surveys are shown with brown boxes and labeled to correspond with the CHIRP profiles shown in Figure 2.

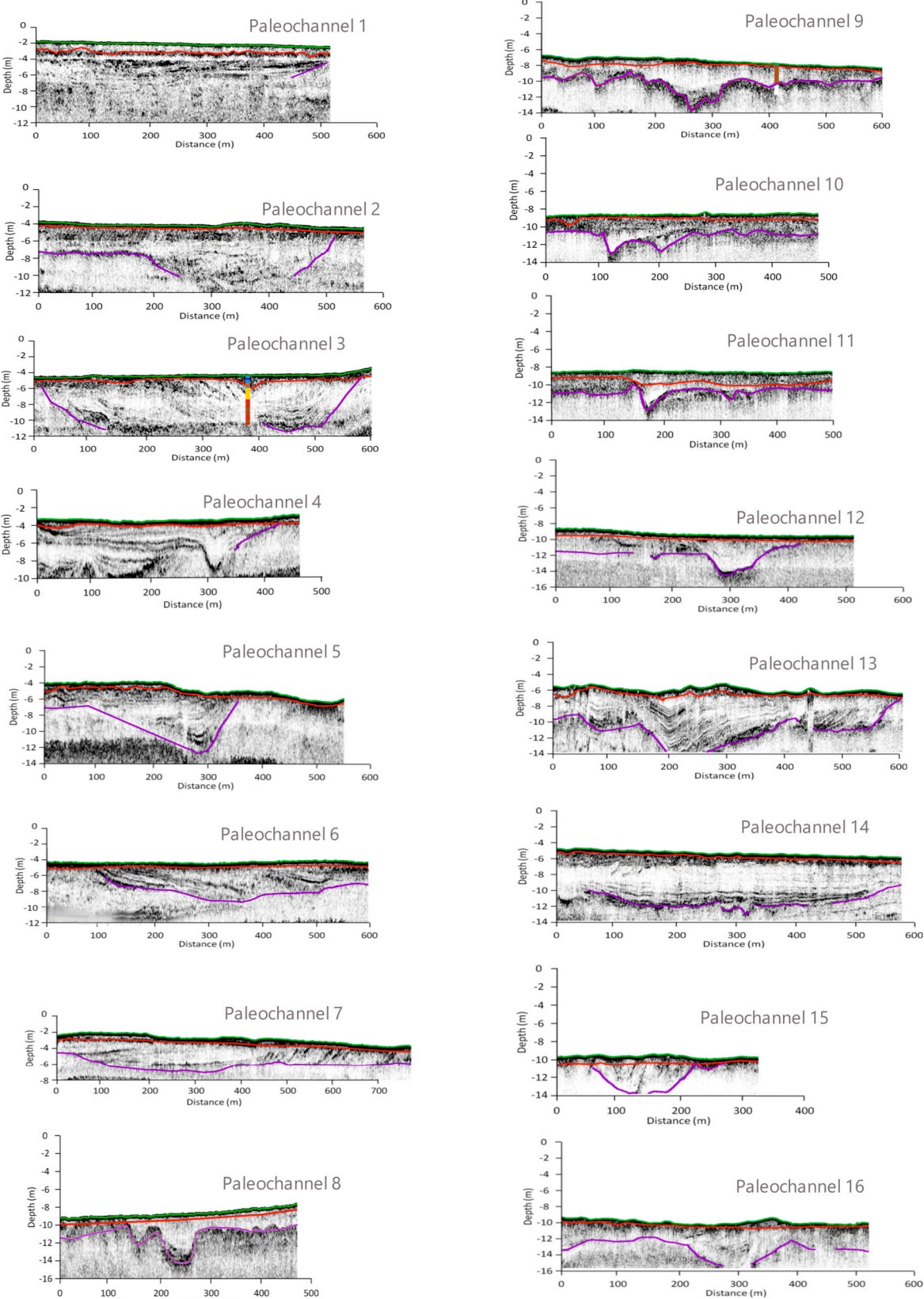

**Figure 2.** Paleochannel crossings from the CHIRP surveys. Each channel crossing is numbered for reference to the location given in Figure 1. The seafloor reflector is a green line, the base of the sand-shell layer is a red line, and the top of the basement is a purple line. Cores are shown in paleochannels 3 (core 1) and 9 (core 5), which correspond to core numbers in Figure 1.

## 1.1. Study Site and Regional Geologic Environment

The study site is a 150 km$^2$ area approximately 2–20 km offshore of the Isle of Palms, which is a barrier island on the north side of Charleston Harbor, South Carolina (Figure 1). This area is ideal to investigate the linkage of sedimentary architecture and SGD because it is easily accessible; previous studies have characterized the underlying pre-Quaternary stratigraphy, regional geologic structures, and bedforms; and it is one of the few where SGD is measured through geochemical tracers and modeling [7,17,34–38].

The study site is located on the Georgia Bight, which is characterized by a wide, shallow continental shelf located on the North American plate. The pre-Quaternary stratigraphy of the Charleston area and the adjacent shelf are fairly established [1,39], but the efforts to characterize the offshore Quaternary sedimentary deposits in this area have been more limited [34,40]. The Quaternary stratigraphy is complex and highly variable regionally [11,32,41]. During the last glacial maximum (ca. 22 ka), the sea level fell to the edge of the continental shelf, and paleochannels were incised into an offshore-thinning Quaternary sediment wedge that is underlain with deposits from the Cretaceous and Cenozoic ages composed primarily of limestone [11,38]. Then, the sea level rose slowly until 16 ka, followed by relatively rapid rise until ~12.8 ka, and it continues rising [42]. This rise in sea-level drowned the previously exposed shelf and deposited transgressive sediments [35,43].

Net longshore transport on the Georgia Bight moves sediment from north to south, but there are local areas of sediment transport reversal caused by tidal inlets and the refraction of waves around ebb-tidal deltas [33–35,44,45]. The Isle of Palms sits at the transition from low aggradation to the north to higher aggradation going south [16,44]. The area is characterized by large ebb-tidal deltas backed by muddy estuaries formed from drowned fluvial valleys or back-barrier tidal creeks. The large tidal inlets, backed by mudflat and tidal creek environments, migrate over time, producing two scales of paleochannels, at the meter scale and at the kilometer scale, in the Quaternary stratigraphy in the same location [11,16,46].

A larger set of paleochannels exist in the Georgia Bight and were mapped by boomer surveys (3.5–5 kHz), and they are on the scale of 1–3 km wide and incise to depths of ~50 m below the seafloor [11,38,43,47–49]. Most studies of these larger paleochannels lack sediment samples due to the water depth, necessary core length, and lack of accurate locations associated with boomer surveys, but a few studies have cores that are not associated with seismic data [11,37,43,46–48]. These studies found that these larger channels contain sand or sandy shell hash deposited in barrier islands and open-shelf environments.

Smaller paleochannels (<1 km wide) are incised in estuarine environments by tidal creeks formed landward of barrier islands in low-energy environments [50]. These smaller channels are not detected on boomer seismic surveys, but a few studies have recently identified these smaller channels in high-frequency seismic records and found laminated mud and silt in their fill [32,38,46,51]. Thieler et al. [46] found more than 2 m of estuarine mud in nearshore channels off of Cape Hatteras, NC, and confirmed that channels farther offshore become sandier as they increase in width and depth. The change in channel fill from sand-dominated large paleochannels to mud-dominated smaller paleochannels suggests that the fill likely varies with paleochannel size and that the continental shelf contained similar features to the current coast throughout the Quaternary.

## 1.2. Previous Studies Relating to Submarine Groundwater Discharge and Paleochannels

Over the last 25 years, hundreds of studies have estimated rates of SGD using a wide range of techniques, ranging from tracer studies, which integrate SGD over the area of interest, to piezometers and seepage meters, which provide information at specific locations [52]. Disconnects can arise because integrative tracers do not indicate where the discharge occurred, and in the absence of geologic information, it is extremely difficult to place piezometers and seepage meters to capture the full range of heterogeneity. Analytical [53] and numerical models (e.g., [12]) have identified the seaward end of confining units

as an important control on SGD, and paleochannels have received significant attention because the cut and fill of paleochannels are one of the main complications to a horizontally stratified "layercake" Quaternary stratigraphy on passive margins (e.g., [14]).

Overall, it has been difficult to determine the effect of geological characteristics on SGD due to a lack of high-resolution geophysical data that help link observations and models, and only a few studies have done so.

Mulligan et al. [17] used a numerical model to examine the effect of a large, symmetrical, high-porosity, and high-permeability paleochannel that breaches a confining layer on SGD. Seismic data were used to estimate the size and shape of the paleochannel, but the reflectors in the channel fill were characterized by large vertical anisotropy that was not accounted for in the flow model. The numerical modeling results indicated that the paleochannel clearly acted as a preferred pathway for the flow of groundwater, and the saltwater–freshwater interface moved slightly landward near a paleochannel [17]. However, these results were local, depending on the permeability of the channel fill and whether the paleochannel breached a confining layer.

In New Jersey, Evans et al. [12] used electrical resistivity as a proxy for zones of fresher pore water and identified paleochannels in seismic data, noting that the resistivity increased only where the paleochannels incised the unconformity between unconsolidated Quaternary sediments and underlying Cenozoic semi-consolidated sediments. In a similar study in Long Bay, SC, Viso et al. [13] recorded highly variable and discontinuous electrical resistivity along lines parallel to the shore with a trend of higher resistivity closer to the shore. Viso et al. [13] found highly irregular "hotspots" of fresher water within 100 m of shore that typically, but not always, correlate with paleochannel incisions.

## 2. Materials and Methods

### 2.1. High-Resolution Seismic Data

High-frequency seismic reflection data were collected over approximately 150 km$^2$ (2.5 × 2.5 km grid) offshore from the Isle of Palms using an Edgetech 3200 sub-bottom system with a 512i towfish from Edgetech, West Wareham, USA. Herein referred to as the 512i, this system uses the Controlled High-Intensity Radar Pulse (CHIRP) method to obtain sub-bottom images of the seabed with vertical resolution of ~10 cm between strata [54]. During our survey, a frequency range of 0.5–10 kHz was used at a towing speed of 4 knots at a constant instrument depth of 1 m below sea surface to maximize sub-bottom penetration with this system. In addition, we used previously published Edgetech 216s CHIRP sub-bottom profiler data acquired a frequency range of 2–10 kHz [33]. The similar frequency range results in comparable sub-bottom resolution, although the 216s has a lower sediment-penetration depth.

We estimate that all of these CHIRP data have a horizontal location uncertainty of 10 m due to the towfish track calculated using the layback navigation from the D-GPS navigation on the boat. The layback navigation was calculated based on the amount of tow cable paid out, the depth of the towfish, and the velocity of the vessel over the water. This is a very common navigation technique and is quite accurate when the towfish is on a line directly behind the tow vessel. All turns between tracklines were removed to reduce navigational error.

The CHIRP data were processed following a typical workflow [55]. The SIOSEIS software version 2015.2.6 [56] was used to filter out the swell, by first picking the seafloor for each trace, and to flatten the seafloor by averaging the seafloor picks over the input estimated oscillation wavelength ranging from 40–80 traces, resulting in an maximum uncertainty of about 1 m in absolute depth. Also in SIOSEIS, a very wide 1–10 kHz bandpass filter was applied to help reduce noise in the water column.

Seismic travel time was converted to depth in ProMax software using an acoustic velocity of 1500 m/s for both datasets for sound in seawater at a shallow depth. Using constant sound velocity introduced uncertainty, which increases with depth from the seafloor because acoustic velocity in sediment is typically somewhat greater than 1500 m/s.

We estimate that the resulting uncertainty was on the order of <1 m at the bottom of the stratigraphic section at ~10 m. Neither tidal variation nor minor changes in instrument depth were able to be corrected, leading to an average vertical uncertainty between track lines of 0.4 m and a range from 0 to 1.6 m based on comparing seafloor depth at crossing lines in the profiles. An Automatic Gain Control filter was also applied at this stage to rescale all of the amplitude values by taking the mean value over a moving window of 10 data samples centered on the adjusted data.

CHIRP data were interpreted using Petrel software to define key reflectors and to tie them to sediment interfaces logged in the cores. Three reflectors were picked in the CHIRP data based on the strength and continuity that would allow them to be carried along the profiles as mostly continuous surfaces. When tied to interfaces identified in sediment cores from each intersection of CHIRP lines, the three reflectors were found to correspond to the following horizons: the seafloor, the bottom of a surficial sand sheet, and the change from the unconsolidated sediment to the semi-consolidated carbonate or marl material (Figure 3). These reflectors were then carried throughout the study area by manually digitizing along the top of the reflector in Petrel. There was an uncertainty in picked horizons based on a maximum distance between the top and bottom of a peak in the wavelength of the CHIRP trace of 0.3 m.

Paleochannels in the CHIRP profiles were defined as concave-up reflectors with minimum dimensions of 200 m horizontally and 0.5 m vertically that define basal erosional surfaces. Features smaller than these were not picked due to the resolution limits of the CHIRP data. Paleochannel fill varies locally and was categorized according to the patterns of reflections within the channel following Long et al. [5]. The meandering of channels across the continental shelf results in oblique intersections of paleochannels in CHIRP profiles. A greater oblique angle yields a higher apparent width, so our measurements represent apparent channel width, usually wider than the true width taken perpendicular to the channel thalweg.

*2.2. Sediment Cores*

Vibracores were taken at the intersection of CHIRP lines and at one additional site in a paleochannel (Figure 1). The coring was conducted by Athena Technologies, Inc., on their custom pontoon vessel using an ~8 cm (3-inch) diameter pipe that was vibrated into the seabed until mechanical refusal. Core locations were determined using D-GPS to correspond to the intersection of CHIRP lines within a 1 m uncertainty radius. Cores were then recovered by winch, cut into 3 m sections, and capped. Once ashore, cores were split down the center and visually inspected and photographed, and one half was sampled at 10 cm intervals for quantitative grain size measurement using a Camsizer particle analyzer at University of South Carolina (grain size data available at osf.io/a4yvx: Core Description.xlsx).

The lithofacies were visually categorized as one of 7 possible categories (Figure 4): (1) shell hash in a poorly sorted sandy matrix, (2) an oyster-rich shell hash in a mud-size matrix, (3) well-sorted arenitic sand, (4) fining-upward sand-to-silt layer with occasional shell fragments, (5) massive, brown-gray mud interbedded with thin layers of silt, (6) massive brown to gray mud, and (7) consolidated to semi-consolidated marl that ranges from muddy sand to mud in grain size.

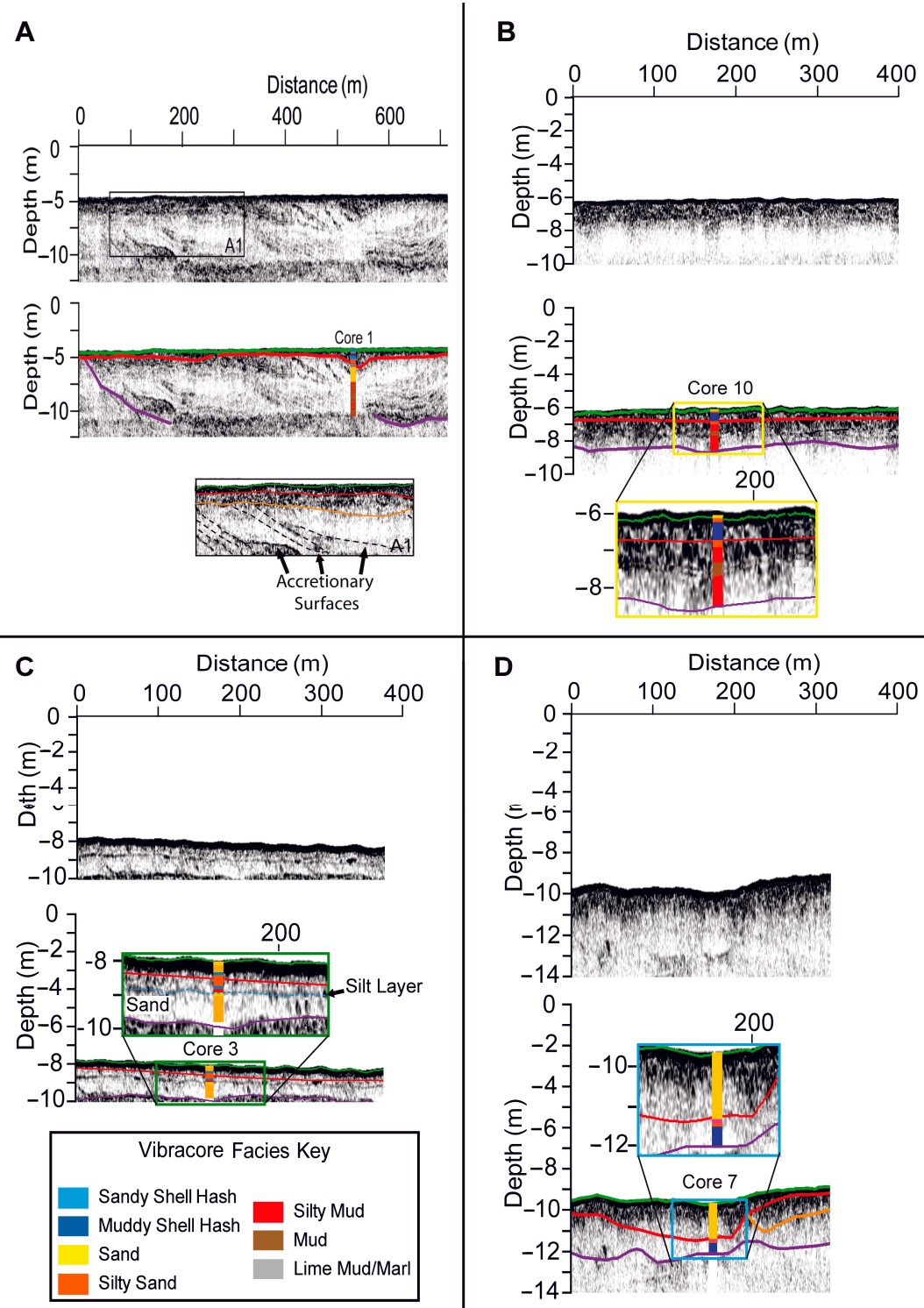

**Figure 3.** Typical 512i CHIRP profiles with the three reflectors used to define seafloor (green), the base of the modern sediment (red) consisting of either sand or shell hash, and the top of the semi-consolidated carbonate basement (purple). An example paleochannel with sand in the upper layers and mixed mud–silty mud layers shows accretionary surfaces (**A**) compared to the typical dappled or chaotic reflectors seen in the top sand-shell hash layers (**B**,**D**). A distinctive light-dappled pattern is occasionally seen below the modern sediment horizon that corresponds with a silty mud layer in cores (**C**).

| Facies name | Photographic example | Sediment color | Grain size | Physical or biological structures | Sorting | Interpreted depositional environment |
|---|---|---|---|---|---|---|
| Sandy shell hash | | Gray-White-Yellow | Medium-coarse | Shells | Poor-Very Poor | Inlet or barrier shoreface |
| Muddy shell hash | | Brown-Gray | fine | Shells, especially oyster | Poor-Very Poor | Backbarrier |
| Sand | | Gray-White | Medium-coarse | -- | Well-Medium | Inlet or barrier island |
| Silty Sand | | Gray-Yellow | Fine-medium | Occasional fining upward | Medium-Very Poor | Inlet or barrier island |
| Silty Mud | | Gray-Brown-Black | Very fine-fine | -- | Medium-Very Poor | Backbarrier or offshore |
| Mud | | Brown | Very fine-fine | Massive, occasional shell | Well-Medium | Backbarrier |
| Marl | | White-Gray | varies | Typically indurated | Well-Poor | Shelf basement |

**Figure 4.** Examples of the 7 distinct lithofacies identified visually from the vibracores with some distinguishing characteristics. Photographic examples taken from the split vibracores are ~5 cm across the frame.

The porosity of each main lithofacies shown in Figure 4 was calculated using a water-displacement method similar to Fraser's [57]. About 300 g of 1 or 2 typical examples of each lithofacies found in the visual lithofacies analysis was collected from cores 1, 2, 5, and 7. The samples were air-dried for at least 24 h and then weighed. The bulk volume of the sample was calculated from the diameter of the half core and the length of the sample. A mortar and pestle were used to unstick grains from one another when necessary. The sample was then placed into a 300 cm$^3$ beaker. Another 300 cm$^3$ of water was measured and poured slowly into the beaker until the sediment was just covered. The sediment was then stirred carefully to remove air from the pores. Water was then poured into the beaker until it was filled to the 300 cm$^3$ mark, and the amount of water left from the original 300 cm$^3$ was recorded as the grain volume. The porosity was then calculated according to

$$\theta = \frac{V_b - V_g}{V_b} \quad (1)$$

where $\theta$ is porosity, $V_b$ is the bulk volume of the sediments, and $V_g$ is the volume of the grains. The porosity of well-sorted sand had a large variance due to one sample of quartz sand and one sample containing a higher fraction of sand-sized shell pieces.

The permeability for each facies was determined using a falling-head test through a permeameter. One sample (~800 g) of each type of lithofacies chosen from the visual litho-

facies analysis was collected from cores 1, 2, 5, and 7 and poured into a permeameter. The permeameter was filled with water under vacuum and allowed to settle for approximately 15 min before water was run through the sample by pouring water in the top standpipe and opening both the top and bottom valves. Once a consistent flow rate was established, the time for a specified volume of water to flow through the sample was recorded. The test was repeated ten times for each sample except mud and silty mud, which were repeated three times. Hydraulic conductivity was calculated as

$$K = \frac{aL}{A\Delta t} \log\left(\frac{h_u}{h_l}\right)$$

(2)

where K is hydraulic conductivity, L is the height of the sample, A is the cross-sectional area of the sample, a is the cross-sectional area of the standpipe, $h_u$ is the head at the top of the standpipe, $h_l$ is the head at the bottom of the standpipe, and $\Delta t$ is the time for the water level to change from $h_u$ to $h_l$. Hydraulic conductivity was then used to calculate permeability as

$$k = K\left(\frac{\mu}{\rho_w g}\right)$$

(3)

where k is permeability, $\mu$ is the viscosity of fresh water at 20 °C, $\rho_w$ is the density of fresh water at 20 °C, and g is the acceleration due to gravity. The variation in the ten iterations for each sample was less than 10% for every sample. The variance of the lime mud was particularly low. This could be because this sample was allowed to settle overnight rather than for only 15 min after being filled with water.

## 3. Results

High-resolution CHIRP seismic-reflection profiles were collected from the inner shelf off Isle of Palms, South Carolina, and interpreted via tie-points at intersecting profiles with sediment layers from cores to develop a stratigraphic framework and improved understanding of the hydrostratigraphy of the site. Three horizons were identified in the CHIRP seafloor data that were tied to the sediment cores and interpreted as the seafloor, the bottom of a surficial sand sheet, and the change from unconsolidated sediment to semi-consolidated marl (Figure 3). A surficial sand sheet is nearly continuous throughout the survey area and consists of the sediment facies categories of silty sand, well-sorted quartz sand, and sandy shell hash. This sediment layer is not a consistent thickness and tends to be thinner offshore, although the general distribution is somewhat patchy overall and varies from 3 m thick to ~0.2 m thick in some sediment cores and is likewise too thin to resolve in the CHIRP profiles.

The prominent reflector between unconsolidated material and consolidated marl was continuous except where it was lost due to signal attenuation or cut by a paleochannel. We refer to this marl layer as the basement, although it is not the true crystalline basement rock, because it marks the base of the unconsolidated sediment. This top-of-marl basement reflector was followed through the seismic data, resulting in thickness values of modern sediment (Figure 5). The general pattern is a thinning of sediment offshore, with the consolidated carbonate directly commonly exposed as hard ground beyond ~11 km offshore. Similar to the surficial sand sediment layer distribution, the total thickness of the sediment above the basement varies locally. Thicker areas are seen closer to the shoreline and in the northeast quadrant of the study area, where paleochannels are most common (Figure 5). Some of the Edgetech 216s profiles lack this basement reflector because the acoustic penetration depth in the higher frequency range was insufficient to image the top of the basement at >7–8 m below the seafloor (Figure 5). Otherwise, the northeast quadrant of our study area consistently has 2–5 m of sediment above the basement.

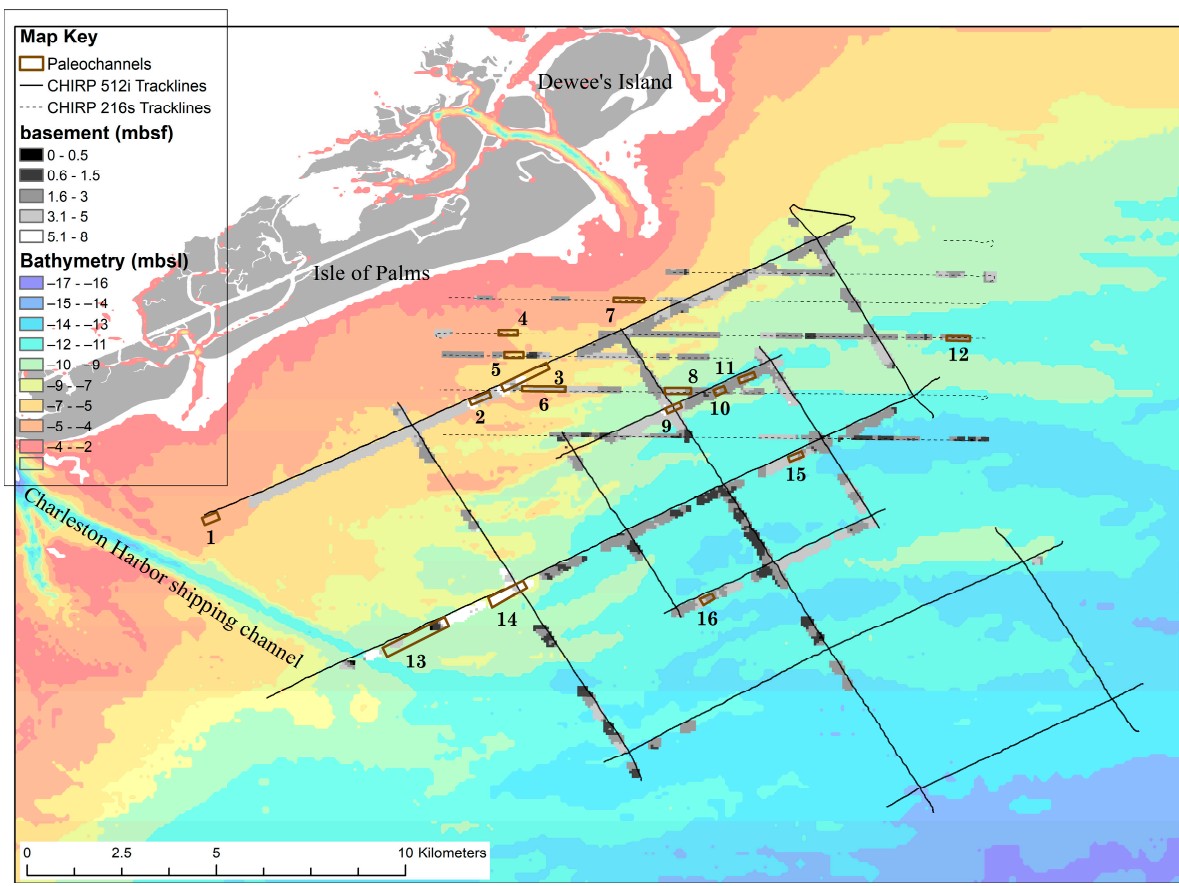

**Figure 5.** Depth to the basement marl in meters below seafloor (mbsf). Darker areas indicate little to no sediment accumulation as the sediment becomes a veneer farther than 11–13 km offshore. Locations where the basement reflector could not be identified in the CHIRP profiles have no color. The brown boxes are paleochannels labeled corresponding to Figure 2.

### 3.1. Paleochannels

Sixteen paleochannel crossings were identified in the CHIRP data based on concave-up reflectors with minimum dimensions of 200 m horizontally and 2 m vertically (Figure 2). Paleochannels were visible on the shore-parallel CHIRP line 10.5 km offshore but were not found on farther offshore lines (Figure 1). No paleochannels were identified on lines perpendicular to the shore, likely due to the unfavorable orientation for imaging a southeast-trending channel. The basal surfaces of the paleochannels ranged in depth from 2.8 m below the seafloor to deeper than the CHIRP penetrated at channel 13, although 13 is unlikely to be much deeper than 9–10 m below the seafloor based on a reasonable extrapolation of the curvature of the channel sides (Figure 2). Paleochannels nearest to the shoreline have their maximum depth at 7–8 m below the seafloor (paleochannels 2–7), and those 2.5 km farther offshore (paleochannels 8–10) are shallower at 5–6 m below the seafloor.

Paleochannels in the study area extend across the continental shelf from 2 km to 11 km offshore but are absent on the shore parallel survey line at 13 km offshore or farther. Their absence in distal locations indicates that the incision was minimal beyond 11 km offshore, suggesting that the fluvial systems approached base level near these locations.

The pattern of CHIRP reflectors can be a useful guide to the cutting of the paleochannels and their subsequent fill, as described in Long et al. [32]. Following their categories, most channels in this study are either the "concentric" or "transparent" types. Paleochannels 1–7 and 13–14 are all concentric with multiple internal CHIRP reflectors that match the curvature of the channel, and channels 8–12 and 16 are transparent without clear internal reflectors (Figure 5). Paleochannel 16 and perhaps paleochannel 2 have a somewhat chaotic

pattern of reflectors. We will address the interpretation of these channel fill patterns in the Discussion (Section 4).

### 3.2. Hydrologic Properties of Sediment Cores

Porosity and permeability were measured for each typical lithofacies type (Table 1). Two types in paleochannels and three from interfluvial sediments were classified for the overall porosity and permeability and for how the porosity and permeability changed with depth (Figure 6). Predictably, the permeability of mud and silty mud was lower than the other facies. The porosity of these unconsolidated muds was also higher than the other lithofacies. The rest of the lithofacies have remarkably similar permeabilities and porosities of roughly $2 \times 10^{-11}$ and 0.45, respectively (Table 1).

**Table 1.** Porosity and permeability of each lithofacies identified in this study.

| Lithofacies | Porosity (%) | Permeability (m$^2$) |
|---|---|---|
| Sandy Shell Hash | 37 | $4.18 \times 10^{-11}$ |
| Well-Sorted Sand | 45 | $1.59 \times 10^{-11}$ |
| Silty Sand | 48 | $1.95 \times 10^{-11}$ |
| Muddy Shell Hash | 47 | $2.12 \times 10^{-11}$ |
| Silty Mud | 68 | $1.20 \times 10^{-13}$ |
| Mud | 67 | $1.06 \times 10^{-13}$ |
| Lime Mud Marl | 43 | $1.56 \times 10^{-11}$ |

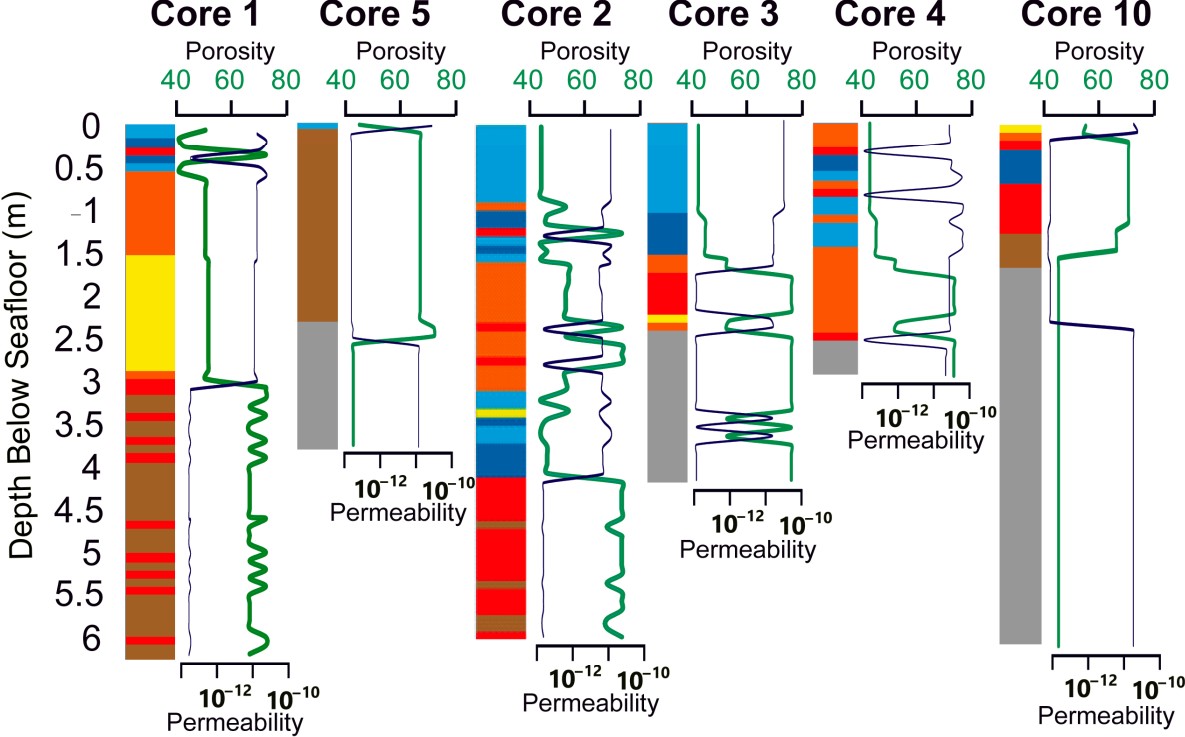

**Figure 6.** Lithofacies, porosity (%), and permeability (m$^2$) logs from selected cores representing paleochannel fill and interfluves. Cores 1 and 5 are from inside paleochannels and recorded generally thicker layers of sediment and a larger concentration of mud than the interfluvial sediments recorded in cores 2, 3, 4, and 10. Facies logs use the color key in Figure 7. Porosity logs are in green, and permeability logs are in purple.

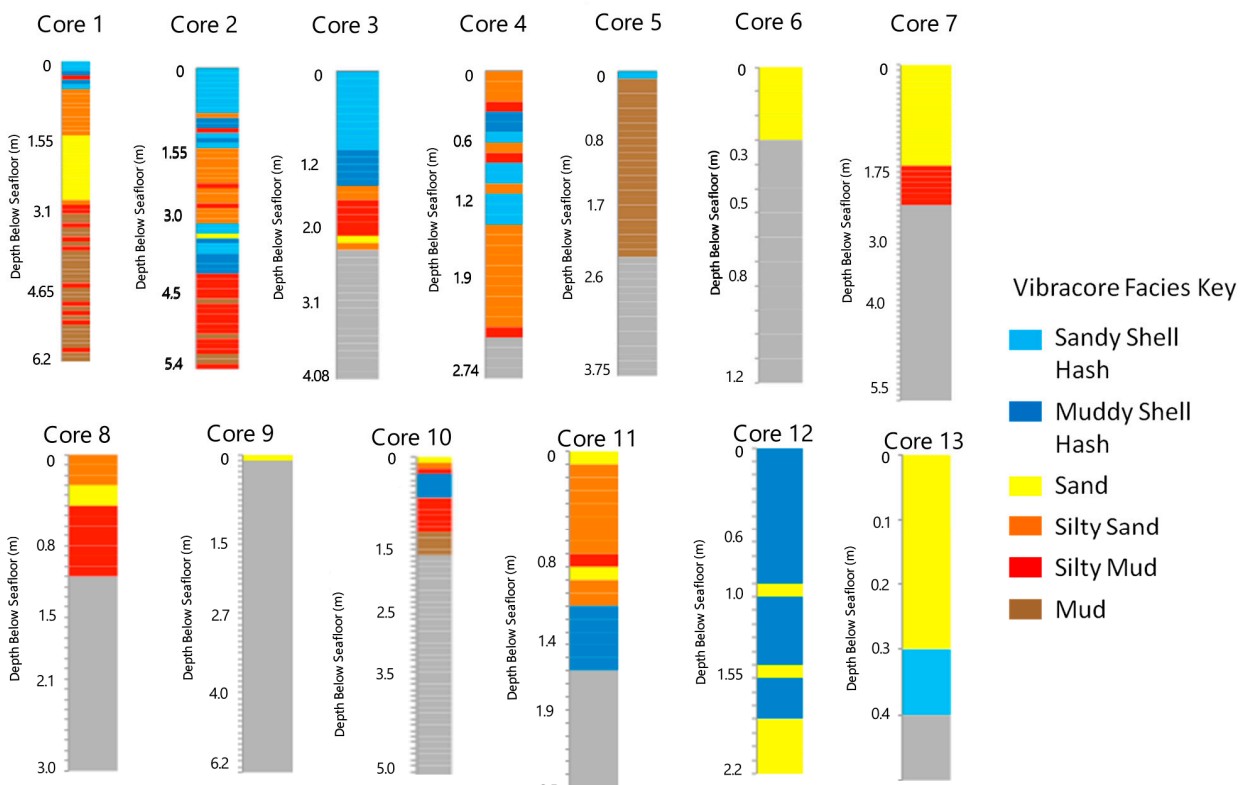

**Figure 7.** Sediment core stratigraphy with depth in each core taken for this study. Facies were determined using visual inspection. Core numbers correspond to the location shown in Figure 1. The gray color indicates consolidated/indurated carbonate marl.

Facies stacking patterns differ between the paleochannels and interfluves. Interfluvial deposits contain thinner layers of sediment and are more variable than paleochannel fill (Figure 6). Interfluvial deposits are typically composed of larger grain sizes, suggesting a higher energy environment during deposition. Many interfluvial deposits also record a mud layer at the top of the unconsolidated sediment column and below the surficial sand sheet. The mud layer is inferred to be widespread across the continental shelf out to ~11 km. The sediment cores in the two paleochannels both show mud to silty mud as the majority of the channel fill in each (Figure 2, paleochannels 3 and 9). Paleochannel fill generally has a higher porosity and lower permeability than interfluvial sediments (Figure 6).

## 4. Discussion

This study documented the presence of a small paleochannel system off the north half of the Isle of Palms (Figure 1). The sediments above the consolidated marl layer are locally thickest in the same area (Figure 5), and CHIRP shows minor incision by the paleochannels (Figure 2). Sediment cores from within and adjacent to the paleochannels contain layers of mud to silty mud that have much lower permeability than any of the other sampled sediments from the area (Table 1). The paleochannels and sediments in our Isle of Palms study area suggest a low-stand environment composed of mud to silty sediments deposited in and around small paleochannels, capped with transgressive-tract sandy layers that are part of a regional system of channels. This general stratigraphic architecture is consistent with other studies of low-accommodation passive margins and very similar to the stratigraphy off of the Santee Delta, 75 km north of our study area [51]. Paleochannels found off Isle of Palms also resemble those documented off of Kiawah Island, south of our study area [32], although those are larger and have a less seismically transparent fill than ours off the Isle of Palms. Most channels in this study are either the "concentric" or

"transparent" types described in Long et al. [32], which they find most commonly indicate muddy fill.

In our study area, paleochannels that contain abundant mud would not be preferential pathways of SGD and probably act instead as part of a confining layer along with the mud layer in interfluvial sediments. Mud-lined or -filled smaller paleochannels in this area corroborate the idea that many smaller paleochannel systems preserve relict estuarine sediment in the Georgia Bight [32]. In the modern environment, Luciano [40] describes a "mud plug" associated with the migration and abandonment of the inlet on Capers Island, immediately northeast of Dewee's Island (Figure 1). We suggest that our area of paleochannels and the Capers Island area represent examples of a back-barrier system preserved on the inner shelf by island migration and sea-level change. The dark brown mud that occupies the paleochannels contains very little recognizable organic material, suggesting that it was deposited in an anoxic, low-energy environment. Similar back-barrier estuarine or marsh settings are present across the U.S. east coast today, where estuarine and tidal-dominated channels are filling with mud-sized sediment [42,49,58]. In addition, we suggest that mud in interfluvial sediments (cores 2, 3, 8, 10) was likely back-barrier intertidal or marsh mudflats between channels because they are slightly siltier than the muds found in the paleochannels (core 1, 5) (Figure 7). The mud layer, like the paleochannels, does not appear farther offshore than 11 km, which could be attributed to the lack of deposition at a paleoshoreline. An alternative possibility is that all paleochannels and back-barrier mud deposits farther offshore were completely eroded during transgression. This latter option is highly unlikely due to the preservation of relict low-stand structures farther out on the shelf within 100 km of the Isle of Palms [35]. These end-member possibilities might be resolved with additional CHIRP and sediment cores that focus on the offshore section of the study area.

The type of sediment filling and surrounding the paleochannels has important implications for understanding SGD. The assumption that paleochannel fill has a high permeability may be valid for larger paleochannel systems or paleovalley systems but is not consistent with our data. The dark brown mud that occupies the paleochannels has a high porosity but very low permeability (Figure 6). Since any sand in the channels is likely surrounded in the subsurface by low-permeability mud, sand in the paleochannels is not likely to be a pathway of fluid flow from the subsurface to the overlying ocean. This suggests alternative interpretations for some previous studies. For example, other geophysical studies found a higher electrical conductivity associated with paleochannels, which suggested the presence of fresher porewater in the channels [12,13]. This could be explained by porewater in a mud layer, which has high porosity but low permeability. Evans et al. [12] suggested that paleochannels that incise contain fresher water, but they noted that salinity variation does not always indicate flow paths.

Broader studies of SGD in the Georgia Bight have shown that the discharge of saline groundwater from the seafloor is similar in volume to river discharge in the region [7,8], and this SGD carries nutrient fluxes that are significant for primary productivity [5]. The mechanism for this discharge was unclear for decades, but recently, pulses of discharge were identified in an offshore well field, using heat as a tracer [9], showing that significant discharge occurs far from shore during persistent wind-driven lows in sea level. The well field identified strong pulses at two sites 15 km from shore but weak or non-existent pulses at sites 10 km from shore. Monitoring less than 10 km from the shore is complicated by shrimping operations, and monitoring more than 15 km from the shore is limited by a lack of unconsolidated sediments for installing the current generation of instrumentation, so it is unclear how much of the seafloor on the broad continental shelf supports this kind of discharge. Marine chemical tracer studies suggest that this discharge can reach as far as 80 km from shore [59], but better knowledge of the Quaternary geological architecture will be necessary to guide future operations.

As previously described, George et al. [9] detected pulses of SGD at wells installed at core sites 7 and 12, 15 km offshore (Figure 1). In contrast, core sites 6, 7, and 9 had little

to no flow at 10 km offshore [9]. The results of our CHIRP and coring, combined with the hydrostratigraphy from the adjacent Charleston region [60], indicate that the groundwater discharge must come from shallow, confined aquifers. The marl basement is the most probable source, as several candidates for this marl layer are laterally continuous onshore [1], and it is a likely aquifer offshore as well [3]. The mechanism driving the flow from this aquifer is a difference in hydraulic head between land and sea level [9]. A confining layer in the Quaternary stratigraphy has not been defined by previous studies in this area, but the interfluvial mud has a low permeability and likely acts as a leaky confining unit for subsurface fluid flow, redirecting the SGD through areas of higher permeability. A confining unit made up of the paleochannels and mud layer may increase the SGD at the edges of the mud layer and between paleochannels. The SGD is most likely to escape in locations that are directly adjacent to the confining layer and are nearest to the present shoreline, which makes defining the areas covered by the mud layer important to understanding SGD flow patterns. In fact, the wells in our study area where the summer pulses of SGD are strongest are the wells just seaward of the paleochannel area [9]. This highlights the multiple scales of groundwater flow that may exist on a typical continental shelf and the need to investigate sites farther offshore, beyond potential mud-confining layers.

## 5. Conclusions

The sediment stratigraphy from 13 vibracores was correlated with high-resolution sub-bottom profile reflectors to interpret the sediment deposits and hydrostratigraphic implications. The upper, unconsolidated sediment generally thins offshore such that the marl may crop out >11 km offshore. A paleochannel system, consisting of small (<9 m deep), simple, and concentric morphology channels, was found offshore of the Isle of Palms that extends from less than 2.5 km offshore to 11–13 km. This suggests that the northern half of the study area contains a relict backbarrier estuarine system. Muds in the study site are high in porosity and low in permeability (Table 1), but the overall sediment accumulation tends to be patchy (Figure 4), resulting in a complex hydrostratigraphy with local confining layers.

This site is part of an integrated study with groundwater monitoring wells, sediment cores, and sub-bottom profiles to better understand the geologic controls on offshore groundwater discharge. The groundwater outflow observed in prior studies at this site was strongest at the monitoring sites just beyond the offshore end of the paleochannels [9] and suggests that the muddy paleochannel system acts as a layer that confines submarine groundwater flow. The correlation of groundwater discharge location and the sediment distribution in our 150 km$^2$ study area may serve as a basis for a better conceptual model that relates submarine paleochannels, the geology, and the locations of submarine groundwater discharge. Such a model may have broader implications for predicting the groundwater flow across the shoreline and farther out on the shelf.

Future studies in the region should not ignore the possibility of groundwater seepage within 11 km from shore, but it is likely that the majority of saline groundwater discharges farther offshore.

**Author Contributions:** Conceptualization, S.M.W. and A.M.W.; methodology, S.M.W. and A.L.L.; validation, S.M.W., E.S. and A.L.L.; formal analysis, E.S.; investigation, E.S., A.L.L. and A.M.W.; resources, S.M.W. and A.L.L.; data curation, S.M.W.; writing—original draft preparation, S.M.W. and E.S.; writing—review and editing, A.L.L. and A.M.W.; visualization, S.M.W.; supervision, S.M.W.; project administration, S.M.W.; funding acquisition, A.M.W. and S.M.W. All authors have read and agreed to the published version of the manuscript.

**Funding:** This research was funded by U.S. National Science Foundation, under grant EAR-1316250.

**Data Availability Statement:** Data generated in this project, Chirp data files and Core descriptions, and photographs are available on Open Science Framework at //osf.io/a4yvx (accessed on 1 July 2023).

**Acknowledgments:** We thank M. S. Harris (College of Charleston), K. Luciano (SC Dept. Nat. Res.) for providing 216s Chirp data, and Coastal Carolina University stafffor help with the 512i CHIRP system. We thank Cory Russell (USC) for assistance with data processing. Suggestions from two anonymous reviewers greatly improved this manuscript.

**Conflicts of Interest:** The authors declare no conflict of interest. The funders had no role in the design of the study; in the collection, analyses, or interpretation of data; in the writing of the manuscript; or in the decision to publish the results.

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
