# Peer review of "Small Muddy Paleochannels and Implications for Submarine Groundwater Discharge near Charleston, South Carolina, USA"

_geosciences, doi:10.3390/geosciences13080232_

Round 1
Reviewer 1 Report
The paper "Small Muddy Paleochannels and Implications for Submarine Groundwater Discharge near Charleston, South Carolina, USA" presents an investigation of sediment deposits and paleochannels in the continental shelf near Charleston, South Carolina, and their implications for submarine groundwater discharge (SGD). The study combines CHIRP seismic mapping with sediment cores to analyze the nature of sediment deposits, including paleochannels, and their relationship to SGD. The authors collected CHIRP seismic reflection data and vibracores in a 150 km2 area and identified eight lithofacies based on grain size, sedimentary structures, and stacking patterns. The results indicate that paleochannels in the study area are often mud-lined or filled, and they coincide with the locations of strong pulses of groundwater discharge. The study seems to be well designed. The methodology and results are technically sound. Discussions on the scientific and practical values of the study, the limitations of proposed models, and future work are meaningful. I recommend accepting this manuscript after revision. The main concerns are as follows:
- The manuscript does not clearly state its research objectives or questions. It is important to explicitly state the purpose of the study and what the authors aim to achieve. This would help provide a clear framework for the research and guide the readers through the manuscript.
- The introduction section does not provide sufficient background information about paleochannels, their significance, and their relationship with groundwater discharge. It is essential to provide a comprehensive overview of the existing knowledge in the field to establish the context for the study.
-Improve the keywords by including only the phrases in the whole body. It is better to avoid using phrases that are repeated in the title.
-Some abbreviations in the paper have already not been addressed in the text.
-I suggest providing a flowchart that illustrates the methodology described in the paper. The methods section would benefit from a more detailed description of the CHIRP seismic data processing, including the specific filtering techniques and criteria used.
-The manuscript would significantly benefit from an expanded literature review encompassing other methods in the related field. I suggest referring to the following sources as examples of contemporary literature in this area:
Hussainzadeh, J., Samani, S., & Mahaqi, A. (2023). Investigation of the geochemical evolution of groundwater resources in the Zanjan plain, NW Iran. Environmental Earth Sciences, 82(5), 123
Samani, S., & Moghaddam, A. A. (2015). Hydrogeochemical characteristics and origin of salinity in Ajabshir aquifer, East Azerbaijan, Iran. Quarterly Journal of Engineering Geology and Hydrogeology, 48(3-4), 175-189.
-The comparison between the current study and previous research could be enhanced by conducting a more extensive literature review.
- The manuscript lacks detailed descriptions of the methods employed in the study. The authors briefly mention the collection of CHIRP seismic data and sediment cores, but they do not provide information about the specific techniques used, sampling locations, or data processing procedures. Providing these details would enhance the reproducibility of the study and allow readers to evaluate the methodology.
- The results section lacks a comprehensive analysis and interpretation of the data. The authors mention the identification of paleochannels, lithofacies classification, and hydrologic properties of sediment cores, but they do not present a thorough analysis or provide meaningful interpretations of these findings.
-The discussion section does not adequately address the implications of the findings. The authors briefly mention the potential implications for submarine groundwater discharge, but they do not elaborate on the broader significance or potential applications of their results. A more in-depth discussion of the implications and their relevance to the existing literature is needed.
- The manuscript does not include a clear and concise conclusion that summarizes the key findings and their significance. A well-defined conclusion is crucial for providing a cohesive end to the manuscript and reinforcing the main points.
Author Response
I have uploaded an attachment with a response to your review and the other review. Thank you for your time spent reviewing this manuscript.

Reviewer 2 Report
SGD, CHIRP etc. Abbreviations must be defined when used for the first time
I am not sure CHIRP must be part of the topic, I guess the particular method of seismics is not the main topic of the work
Lines 30-31 “Paleochannels are common features within modern continental shelf environments, particularly on the southeastern United States” not “particularly”, but “also”, or “as also found on SE U.S.” ro something that kind
Line 39: what is “traer”?
Line 66: “ideal” for what? Did the authors mean “selected because of …”?
Line 75 and elsewhere: what is “kya”? I met “ky” (“kyr”) and “ka” (with a different meaning), but not “kya”
Lines 151-152: “…to guide future operations during the Quaternary” do the authors plan to continue some studies later in Quaternary? I do not understand that text.
Line 158: space between numbers and unites everywhere
Line 159: “KHz” is wrong, it should be “kHz”
Line 175: reference to/description of method is missing
Section 2.2: the way of getting cores must be specified!! I did not find info whether the cores were available from preceding work, or obtained particularly for this manuscript and yet not reported, I have no idea how coring is performed below the water column. Perhaps real section 2.2 “Sediment Cores” is missing, while the text actually below that heading should be something like 2.3 “Sediment Characterization”
Line 245 and farther: I would recommend to put the characteristics of the 8 types of facies to a table and integrate Fig. 3 into this table. Lithofacies should be somehow clearly labelled and the labels used in the manuscript, e.g. in Table 1 and Fig. 7
Lines 289-290 and elsewhere, plots in Fig. 6. Units should be specified
Table 1: what is “sandy shell” or “muddy shell”? Elsewhere in the manuscript, the authors use “sandy shell hash” or “sandy shell layer”, but in Table the noun seems missing
Conclusion. I do not consider use of abbreviations in this section reader-friendly.
This section summarises the findings reported by authors, findings from other sources (references) should not be placed here.
Author Response
Please see the attached response to your review and the other reviewer's concerns. Thank you for your time, and I hope that my revisions do justice to address your review.

Round 2
Reviewer 1 Report
Paper can be accepted.